

# Revising the definition of anthropogenic heat flux from buildings: role of human activities and building storage heat flux

Yiqing Liu [1], Zhiwen Luo[1], Sue Grimmond[2]

[1] School of the Built Environment, University of Reading, Reading, UK

[2] Department of Meteorology, University of Reading, Reading, UK

*Correspondence to*:  * Zhiwen Luo (z.luo@reading.ac.uk) and Sue Grimmond (c.s.grimmond@reading.ac.uk)

**Abstract.** Buildings are a major source of anthropogenic heat emissions, impacting energy use and human health in cities. The difference between building energy consumption and building anthropogenic heat emission magnitudes and time lag and are poorly quantified. Energy consumption ($Q_{EC}$) is a widely used proxy for the anthropogenic heat flux from buildings ($Q_{F,B}$). Here we revisit the latter's definition. If $Q_{F,B}$ is the heat emission to the outdoor environment from human activities within buildings, we can derive it from the changes in energy balance fluxes between occupied and unoccupied buildings. Our derivation shows the difference between $Q_{EC}$ and $Q_{F,B}$ is attributable to a change in the storage heat flux induced by human activities ($\Delta S_{o-uo}$) (i.e., $Q_{F,B} = Q_{EC} - \Delta S_{o-uo}$). Using building energy simulations (EnergyPlus) we calculate the energy balance fluxes for a simplified isolated building (obtaining $Q_{F,B}$, $Q_{EC}$, $\Delta S_{o-uo}$) with different occupancy states. The non-negligible differences in diurnal patterns between $Q_{F,B}$ and $Q_{EC}$ caused by thermal storage (e.g. hourly $Q_{F,B}$ to $Q_{EC}$ ratios vary between -2.72 and 5.13 within a year in Beijing, China). Negative $Q_{F,B}$ can occur as human activities can reduce heat emission from building but are associated with a large storage heat flux. Building operations (e.g., open windows, use of HVAC system) modify the $Q_{F,B}$ by affecting not only $Q_{EC}$ but also the $\Delta S_{o-uo}$ diurnal profile. Air temperature and solar radiation are critical meteorological factors explaining day-to-day variability of $Q_{F,B}$. Our new approach could be used to provide data for future parameterisations of both anthropogenic heat flux and storage heat fluxes from buildings. It is evident that storage heat fluxes in cities may also be impacted by occupant behaviour.

## 1 Introduction

Human's activities that influence energy exchanges are critical to a wide variety of disciplines (e.g. meteorology, building design, geography, climatology, hydrology, engineering). As disciplines often have interests in different scales, purposes and/or boundary conditions, the terminology and acceptable assumptions



differ. However, disciplines may provide data to each other or help improve assumptions used. In this study we
are concerned with the interface between meteorology, climatology and building design in urban areas.
To model the weather and climate in urban areas, an important additional source of energy to the environment is
the anthropogenic heat flux ($Q_F$). This is defined as the heat converted from consumption of biological,
chemical and electrical energy and released to the atmosphere due to human activities (Oke et al., 2017). $Q_F$ has
three major sources, including metabolic (people and animals) activities ($Q_{F,M}$), transport ($Q_{F,T}$) and buildings
($Q_{F,B}$) (Grimmond, 1992). It can be large relative to incoming solar radiation in summer (e.g. 43% in an area of
Beijing (Nie et al., 2014)) and increases air temperature in cities (e.g.(Ichinose et al., 1999; Fan and Sailor,
2005)), subsequently contributing to higher cooling demand for buildings (Santamouris et al., 2001; Takane et
al., 2019). Apart from that, $Q_F$ is also a dominant attribution of wintertime urban heat island (Biggart et al.,
2021).Compared to $Q_{F,M}$ and $Q_{F,T}$, the generated heat within building volume is not all directly ejected into the
outdoor environment. For example, the heat from mechanical heating system is released in the indoor
environment, then conducted into building fabric and eventually emitted into atmosphere through sensible
turbulent heat flux and outgoing longwave radiation. In this process the net storage heat flux ($\Delta Q_S$) of building
is modified since building fabric temperature is changed by mechanical heating system with absorbing more
heat.

In urban areas, $\Delta Q_S$ is the net uptake or release of energy from urban volume. This term is an important

determinant of urban climate and is regarded as a key process in the genesis of urban heat island (Goward,
1981). The change in building $\Delta Q_S$ is modified when heat is released by human activities but the timing of the
externally emissions are impacted by the building fabric characteristics and the conduction process. With prior
studies often using energy consumption ($Q_{EC}$) as a proxy for $Q_{F,B}$ from inventory related approaches (e.g. Sailor
and Lu, 2004; Iamarino et al., 2012) and building energy modelling (e.g. Heiple and Sailor, 2008; Nie et al.,
2014) , the impact on $\Delta Q_S$ is not addressed. To qualify the 'real' $Q_{F,B}$ and change of $\Delta Q_S$, we revisit the
definition of $Q_{F,B}$ and attempt to understand how human activities affect the energy balance fluxes of building.

If $Q_{F,B}$ is the heat released from buildings into the atmosphere as a result of human activities inside the

building (including human metabolism), when the building is completely unoccupied (e.g. no operational
appliances, no people) $Q_{F,B}$ is zero. However, heat released from the unoccupied building is non-zero as there is
still heat exchange between building and ambient environment. Shortwave and longwave radiation can enter the
unoccupied internal building space through windows and conduction through walls. This energy modifies the
internal building volume, influencing storage heat flux and the other terms of the energy balance. These are not


anthropogenic heat flux when the energy leaves the unoccupied building but influence the heat emissions from
the building. This is consistent with radiation penetrating deep into water, and similarly allowing a larger
volume to be heated than soil because of convection (Sellers, 1965).

For an occupied building, the internal heat gain arises from: (1) the equivalent sources and sinks as the

unoccupied buildings; but also (2) the energy linked to the indoor human activities (metabolism, powered
appliances and energy inputs to heating or cooling). These will modify each of the energy balance fluxes. Some
of this additional energy is transported out of buildings through indoor-outdoor ventilation exchange and
immediately contributes to $Q_{F,\mathrm{B}}$, while some is stored in the building fabric, and later released outdoors through
various pathways (convection, radiation, conduction) to become $Q_{F,\mathrm{B}}$ with a time lag. Here, we will derive
$Q_{F,\mathrm{B}}$ by looking at the difference of heat fluxes between occupied and unoccupied buildings.

If the energy balance for the building system (including the indoor air and building envelope) for an

unoccupied dry building (assuming latent heat is not important in this case) is:
$$Q_{uo}^* = Q_{H,\mathrm{uo}} + Q_{BAE,\mathrm{uo}} + \Delta Q_{S,\mathrm{uo}} \tag{1}$$
The radiation balance for an isolated unoccupied (*uo*) building can be expressed as:
$$Q_{\mathrm{uo}}^* = K_{\downarrow,\mathrm{uo}} - K_{\uparrow,\mathrm{uo}} + L_{\downarrow,\mathrm{uo}} - L_{\uparrow,\mathrm{uo}} \tag{2}$$
where $Q^*$ is the net all-wave radiation, K is the shortwave radiation incoming ($\downarrow$) and outgoing ($\uparrow$) to the
external surfaces. The longwave (L) radiation exchanges depend on the view factors (F) between the building of
interest (*boi*), the surrounding facets of other surfaces/buildings (*other b*) and the sky:
$$L_{\downarrow,\mathrm{uo}} = L_{\downarrow,\mathrm{uo}(F[sky \rightarrow boi])} + L_{\downarrow,\mathrm{uo}(F[other\ b \rightarrow boi])} \tag{3}$$
$$L_{\uparrow,\mathrm{uo}} = L_{\downarrow,\mathrm{uo}(F[boi \rightarrow sky])} + L_{\uparrow,\mathrm{uo}(F[boi \rightarrow other\ b])} \tag{4}$$

In Eq. (1), $Q_H$ is the turbulent sensible heat flux (convection) from external surfaces to the external ambient

air. $Q_{BAE}$ is the net energy exchange from the buildings through air exchange (e.g. ventilation). When the
building is sealed $Q_{BAE}$ is 0 W m$^{-2}$, otherwise (e.g. open windows, cracks) it can be a source or sink of energy
(environment ← building, or inverse). $\Delta Q_S$ is the net storage heat flux of the building volume (i.e. fabric,
contents, including the air). The left-hand side (LHS) of Eq. (1) is the inputs or source of energy to the building,
whereas the right-hand side (RHS) is the sink or energy dissipation outputs. With no human activities within the
building and the internal heat generation from human and infrastructure activities is zero.

When the building is occupied (*o*) (e.g. appliances operating) additional terms are needed in Eq. (1) to

account for the supply of energy into the building for these activities and the release of energy:
$$Q_{\mathrm{o}}^* + Q_{Internal,\mathrm{o}} + Q_{HVAC,\mathrm{o}} = Q_{H,\mathrm{o}} + Q_{BAE,\mathrm{o}} + \Delta Q_{S,\mathrm{o}} + Q_{Waste,\mathrm{o}} \tag{5}$$
The two additional sources of energy (LHS) are:
(1) $Q_{Internal,o}$ : energy released within the building from lighting, powered appliances and metabolism (e.g.
people, pets).
(2) $Q_{HVAC,o}$ : energy consumption in the building from heating, ventilation and air conditioning (HVAC) system.
As the building may emit exhaust/waste heat (e.g. via HVAC systems), there is an additional sink (RHS)
referred to here as $Q_{Waste,o}$.
To determine the impact of the occupancy (i.e. not just the physical building form) we can consider the
difference between Eq. (5) and Eq. (1). If the radiation balance for the occupied case is:
$Q_o^* = K_{\downarrow,o} - K_{\uparrow,o} + L_{\downarrow,o} - L_{\uparrow,o}$         (6)
We assume that the incoming and outgoing shortwave radiation remains unchanged because the reflectivity,
transmissivity and absorptivity do not change by occupancy activities then:
$K_{\downarrow,o} = K_{\downarrow,uo};$       $K_{\uparrow,o} = K_{\uparrow,uo}$
The incoming longwave radiation is dependent on the surroundings which are independent to the building state,
so:
$L_{\downarrow,o} = L_{\downarrow,uo}$
Thus, the difference of radiative fluxes between occupied and unoccupied building ($\Delta L_{\uparrow,o-uo}$) is:
$\Delta L_{\uparrow,o-uo} = L_{\uparrow,o} - L_{\uparrow,uo}$         (7)
Similarly, the difference of the heat transfer through air exchange is:
$\Delta BAE_{o-uo} = BAE_o - BAE_{uo}$         (8)
With the additional terms in Eq. (5) and the air exchanges rates difference from the activities within the
buildings, gives:
$\Delta B_{o-uo} = [Q_{Internal,o} + Q_{HVAC,o}] - [Q_{Waste,o} + \Delta BAE_{o-uo}]$         (9)
As the change in surface temperature influences the sensible heat fluxes and storage heat fluxes:
$\Delta H_{o-uo} = H_o - H_{uo}$         (10)
$\Delta S_{o-uo} = \Delta Q_{S,o} - \Delta Q_{S,uo}$         (11)
By combining the Eq. (1) and Eq. (5), we obtain:
$\Delta B_{o-uo} = \Delta L_{\uparrow,o-uo} + \Delta H_{o-uo} + \Delta S_{o-uo}$         (12)
where the LHS accounts for the net available energy as result of human activities in indoor environments and
the RHS shows that these impact the longwave radiation, turbulent sensible and storage heat fluxes (in this dry
case). With rearrangement:


$$\left[Q_{Internal,o} + Q_{HVAC,o}\right] = \Delta S_{o-uo} + [\Delta L_{\uparrow,o-uo} + \Delta H_{o-uo} + \Delta BAE_{o-uo} + Q_{Waste,o}] \tag{13}$$
The additional energy generation associated with human activities to the whole building system (LHS) is
apparent, as traditionally defined as $Q_{F,B}$ previously (Heiple and Sailor, 2008). Here because the heat release
from human metabolism indoors is considerably smaller than other sources, for simplicity of analysis, we
assume metabolic heat is also part of energy consumption ($Q_{EC} = Q_{Internal,o} + Q_{HVAC,o}$). Besides, some of
additional energy is associated with the extra gain or release of stored heat within the building volume ($\Delta S_{o-uo}$).
The rest is the heat released to outdoor environment from building due to human activities, which is the $Q_{F,B}$
based on its definition:
$$Q_{F,B} = \Delta L_{\uparrow,o-uo} + \Delta H_{o-uo} + \Delta BAE_{o-uo} + Q_{Waste,o} \tag{14}$$
Eq. (14) demonstrates the $Q_{F,B}$ is the relative heat emission at exterior building boundary between
unoccupied and occupied building through longwave radiation, convection, air exchange and waste heat from
mechanical heating/cooling system. The source of $Q_{F,B}$ within the building volume gives (by combining Eq.
(13) and Eq. (14):
$$Q_{F,B} = Q_{EC} - \Delta S_{o-uo} \tag{15}$$
The sources of $Q_{F,B}$ are from both energy consumption ($Q_{EC}$) and difference of storage heat flux ($\Delta S_{o-uo}$)
between unoccupied and occupied building ($Q_{F,B}$ in this study includes part of $Q_{F,M}$ from human metabolism).
Whereas the second term is ignored in most prior studies and consequently leads to a time lag and magnitude
difference between $Q_{F,B}$ and $Q_{EC}$ (Sailor, 2011). Therefore, estimation of $Q_{F,B}$ by differences in heat emission
between occupied and unoccupied building can capture impact of dynamic changes in the building storage heat
flux.
In this study, the objective is to understand the temporal profile of $Q_{F,B}$, and how and why it differs from
$Q_{EC}$ at diurnal and seasonal time scales, by examining differences in energy balance fluxes between an occupied
and unoccupied same building. Building energy simulation tool (EnergyPlus) is used to obtain the various
energy balance fluxes from the building system.
**2 Methods**
**2.1 Unoccupied (uo) and occupied (o) building energy simulation (BES)**
Building energy simulation (BES) is widely used to estimate energy consumption, heat emission and heat
storage within a building, while allowing changes in heat fluxes due to human activities to be estimated. Here





we use EnergyPlus version 9.4 (DOE, 2020) to study an isolated building (i.e. without a surrounding
neighbourhood). The ASNI/ASHRAE standard 140 Case 900 test model (ASHRAE, 2017) is used, which is
developed in a software-to-software comparative tests for validating building thermal load. It is a 48 m$^2$ one-
story heavyweight rectangular prism with high mass fabrics (Appendix A), whose simple geometry is ideal to
understand the process of how human activities change the building energy balance fluxes in a theoretical study.
Modifications of the original building model for this study, include: windows are reduced to one (6 m$^2$ south-
facing) for more appropriate EnergyPlus single-sided ventilation calculations (Daish et al., 2016); and internal
heat gain, ventilation control strategy and HVAC system operation vary with different scenarios considered
(Table 1). For the simulations, the building is assumed to be located in Beijing as the climate has both hot
summer and cold winter conditions. Chinese Standard Weather Data (CSWD) selected to create a Typical
Meteorological Year (TMY) (China Meteorological Bureau et al., 2005) are used as the meteorological forcing,
as these data are developed for simulating building thermal load and energy use.

The modelling scenarios (Table 1) vary with building occupation state. Two types of unoccupied (*uo*)

buildings are considered. Neither have internal heat gains nor HVAC systems, but they differ based on air
exchange between (1) unoccupied sealed (*us*) with no infiltration or ventilation, and (2) unoccupied ventilated
(*uv*) with 50% of windows area kept open. The single-sided natural ventilation rate is estimated by including
both wind-driven ventilation rate ($V_W$, m$^3$ s$^{-1}$) (Warren 1977):
$V_W = 0.025 A_{eff} U_W$ (16)
and the stack buoyancy-driven ventilation rate ($V$, m$^3$ s$^{-1}$) (Warren 1977):
$V_{Stack} = \frac{1}{3} A_{eff} C_d \sqrt{\frac{\Delta T H g}{T_{ave}}}$ (17)
where $A_{eff}$ is the effective opening area (m$^2$), $U_W$ is reference wind speed at the height of opening (m s$^{-1}$). $C_d$
is discharge coefficient (usually taken as 0.6 (Wang and Chen, 2012)), $\Delta T$ is indoor and outdoor air temperature
difference (℃), $H$ is the height of opening (m), $g$ the gravitational acceleration (m s$^{-2}$), $T_{ave}$ is average indoor
and outdoor air temperature (℃). The combined ventilation rate is (Fan et al., 2021):
$V_T = \sqrt{V_W^2 + V_{Stack}^2}$ (18)

The three occupied (*o*) building simulations assume occupant behaviour modifies internal heat generation,

natural ventilation and HVAC systems (*ov*). First, *ov1* has internal heat gains ($Q_{Internal,o}$) from human
metabolism, lighting and other appliances based on local building code (MOHURD, 2018), with window always





open (50%, as $uv$). The internal heat gains are held constant allowing the fraction of heat in $Q_{F,B}$ and $\Delta Q_S$ to be
impacted by building and climate conditions but not the diurnal variability of human heat generation.
Second, $ov2$ considers natural ventilation based on passive cooling and thermal comfort. The window
opening is controlled automatically. It is opened (50% of window area) when the indoor air temperature is
higher than both outdoor air temperature and ventilation setpoint (23℃ for 'warm limit' in bedroom
(Oikonomou et al., 2012)). Otherwise, it is closed to reduce heat loss and keep the building warm. Third, since
natural ventilation alone may not satisfy indoor thermal comfort, mixed mode ventilation with auxiliary HVAC
system (e.g. Wang and Chen, 2013; Wang and Greenberg, 2015; Chen et al., 2017) is considered in $ov3$. The
mechanical heating and cooling system are active when indoor temperature reaches the threshold (18℃ for
heating and 26℃ for cooling, MOHURD, 2018). The ventilation control strategy in $ov3$ is the same as $ov2,$ but
the EnergyPlus hybrid ventilation manager (DOE, 2020) turns the HVAC off when natural ventilation is active
to prevent simultaneous operation.
Table 1. Cases simulated differ based on building occupation state, internal heat gain ($Q_{Internal,o}$ ) and presence of natural
ventilation and HVAC. Notation are defined in text and nomenclature

| Code | Occupation state | Natural ventilation | $Q_{Internal,o}$ (W m⁻²) | Window open Temperature control (°C) | HVAC Heating/ cooling setpoint (°C) |
|---|---|---|---|---|---|
| us | uo | Sealed | 0 | N/A | N/A |
| uv | uo | Window always open (50%) | 0 | N/A | N/A |
| ov1 | o | Window always open (50%) | 11.8 | N/A | N/A |
| ov2 | o | Controlled ventilation | 11.8 | 23 | N/A |
| ov3 | o | Mixed mode control | 11.8 | 23 | 18/26 |

**2.2 Determination of anthropogenic heat flux**
The simulated hourly heat fluxes by radiation, convection, air exchange and waste heat generated from HVAC
system between the isolated building and atmosphere (Table A.3) are analysed for each case (Table 2). If
cooling occurs, the waste heat consists of the cooling load and electrical energy consumed by the air conditioner
($Q_{HVAC}$). $Q_{HVAC}$ is predicted using a static coefficient of performance (COP) for the air conditioner, and the heat
removed by an air conditioner ($Q_{AC}$) to the total amount of electricity consumed:
$Q_{HVAC,C} = \dfrac{Q_{AC}}{COP}$                                                                 (19)
$Q_{Waste,C} = Q_{AC}(1 + COP^{-1})$                                                      (20)
With a centralised heating system (as Beijing has), for simplicity we assume all energy associated with the
heating system is released indoors, and waste heat due to boiler efficiency and pipe heat loss are not considered:
$Q_{HVAC,H} = Q_{HS}$                                                                          (21)



$Q_{Waste,H} = 0$ $\hspace{12em}$ (22)
Combing these, and accumulated though time gives annual values:
$Q_{HVAC} = Q_{HVAC,C} + Q_{HVAC,H} = \frac{Q_{AC}}{COP} + Q_{HS}$ $\hspace{8em}$ (23)
$Q_{Waste} = Q_{Waste,C} + Q_{Waste,H} = Q_{AC}(1 + COP^{-1})$ $\hspace{6em}$ (24)
$\qquad$ Each term in Eq. (14) is determined using an occupied (*o*) and unoccupied (*uo*) building result to determine
$Q_{F,B}$ and the other fluxes. The results are analysed by season (spring (March, April and May; MAM), summer
(JJA), autumn (SON) and winter (DJF)) using the median (50%) and interquartile range (IQR) between the 25[th]
and 75[th] percentiles to assess the diurnal patterns.
**2.3 Ratio of anthropogenic heat flux to energy consumption**
If the energy consumed within the building is rejected immediately into the atmosphere (Heiple and Sailor,
2008), the change in $\Delta Q_S$ is not accounted for, and therefore $Q_{F,B}$ is assumed to be only from energy
consumption ($Q_{EC}$). The variation of $\Delta Q_S$ associated with human activities is considered when using the relative
heat emissions in Eq. (14) and Eq. (15). We use the ratio $R = \frac{Q_{F,B}}{Q_{EC}}$ to determine the relative importance of
building operation modes and choice of baselines on the discrepancy between $Q_{F,B}$ and $Q_{EC}$.
**3 Results and discussion**
Building energy balance fluxes vary through each day and season (Fig. 1) associated with when a building is
occupied and people's activities inside the building. First, we consider one case in detail - an occupied building
with both natural ventilation and HVAC (*ov3,* Table 3) relative to an unoccupied sealed building (*us,* Table 4) -
their difference (*ov3-us*) allows us to obtain the fluxes needed (Sect. 1).
$\qquad$ As noted (Sect. 1), the shortwave and incoming longwave radiative fluxes for all cases (Table 5) are
assumed identical, but all other terms of the building energy balance differ. Hence, the change in outgoing
longwave radiation ($\Delta L_{\uparrow,o-uo}$, Fig. 1c) is equivalent to the net all-wave radiation difference ($Q^*_{o-uo}$, Fig. 1a-b)
for the occupied and unoccupied buildings. The positive sensible heat flux difference (Eq. (10), $\Delta H_{o-uo}$, Fig. 1c)
and $\Delta L_{\uparrow,o-uo}$ indicate the building is warmed up by internal heat gains ($Q_{Internal,o}$) with higher exterior surface
temperatures. Their small magnitudes and flat patterns indicate small relative importance compared to the heat
exchange from ventilation differences (Eq. (8), $\Delta BAE_{o-uo}$, Fig. 1c). The latter, not only contributes the largest
fraction of anthropogenic heat flux ($Q_{F,B}$, Fig. 1c), but also has a diurnal pattern consistent with $Q_{F,B}$, especially



during spring and autumn (Fig. 1c, i). Rarely, heat ($Q_{Waste,o}$, Fig. 1i) is emitted by the air conditioner in the
mid-afternoon (shading) at this time of year, but more importantly in summer (Fig. 1f) when cooling demand
increases.

$Q_{F,B}$ (Eq. (14), Fig. 1c) has four components of emitted heat, whereas energy consumption ($Q_{EC}$, Fig. 1c)

only has (in this case, constant) internal heat gains ($Q_{Internal,o}$ = 11.8 W m$^{-2}$, Fig. 1b, Table 6) and energy use
from HVAC system ($Q_{HVAC}$, Fig. 1b). Their difference is the storage heat flux difference (Eq. (15) $\Delta S_{o-uo}$in Fig.
1c). If $\Delta S_{o-uo}$ is positive, the building acts as a heat sink and stores the extra heat generated by human activities,
or stored heat is released when $\Delta S_{o-uo}$is negative. Hence, we can identify the impacts of seasonal-varying
human activities and building operations on the diurnal variability in $\Delta S_{o-uo}$, $Q_{EC}$ and $Q_{F,B}$.




Figure 1: Seasonal diurnal median (line) and inter-quantile range (IQR, shading) building heat fluxes for (a, d, g, j)

unoccupied sealed (us), (b, e, h, k) occupied ventilated (*ov3*) building and their (c, f, i, l) difference (*ov3-us*) for (a-c) spring,

(d-f) summer, (g-i) autumn and (j-l) winter. $Q_{F,B}$ is estimated by either heat transfer difference (solid line components):

$Q_{F,B} = \Delta L_{\uparrow,o-uo} + \Delta H_{o-uo} + \Delta BAE_{o-uo} + Q_{Waste,o}$ in Eq. (14) or energy consumption and storage flux difference: $Q_{F,B} =$

$Q_{EC} - \Delta S_{o-uo}$ (dash line components) in Eq. (15)




**3.1 Impact of human activities on seasonal and diurnal variations of the fluxes**

For the same *ov3-us* case (Table 1, Fig. 1), we consider the temporal and seasonal variability of the fluxes. In spring and autumn (Fig. 1a-c, g-i), natural ventilation is the dominant factor contributing to diurnal variation in $\Delta S_{o-uo}$ and $Q_{F,B}$, while $Q_{EC}$ has minimal variability. $Q_{EC}$ is slightly larger than $Q_{Internal,o}$ because of some short periods of HVAC use in the mid-afternoon (IQR shading in Fig. 1i). There is a clear diurnal cycle of $Q_{F,B}$ (Fig. 1c) with the median varying between 8 W m$^{-2}$ (07:00) and 15 W m$^{-2}$ (15:00) relative to the constant internal heat gain (11.8 W m$^{-2}$). The difference between $Q_{F,B}$ and $Q_{EC}$ ($\Delta S_{o-uo}$) is largely impacted by natural ventilation. During the night and early morning with closed window, only part of the consumed energy is transferred externally to the atmosphere. The rest of the heat is stored in the building fabric (positive $\Delta S_{o-uo}$), hence $Q_{F,B}$ is lower than $Q_{EC}$. However, when overheating may occur during the middle of the day, occupants keep window opened (air conditioner is less frequently used) to cool the building down, with stored heat released (negative $\Delta S_{o-uo}$). This is consistent with the diurnal variability of $\Delta BAE_{o-uo}$, which has a minimum at night (window closed) and maximum in the mid-noon (window open).

In summer, the role of natural ventilation at daytime is replaced by air conditioning. Natural ventilation and waste heat from the air conditioner ($Q_{Waste,o}$) contribute to one peak $Q_{F,B}$ at nighttime and daytime, respectively (Fig. 1f). $Q_{F,B}$ is higher than $Q_{EC}$ around these two peak periods (05:00-07:00 and 13:00-21:00). The peak $Q_{F,B}$ at night reaches 14 W m$^{-2}$ (median) at 05:00, which is mainly attributed to natural ventilation when outdoor air temperature is cooler than indoors. Conversely, in the afternoon when outdoor temperature is warmer, occupants 'choose' mechanical cooling for achieving thermal comfort. The peak $Q_{F,B}$ is 22 W m$^{-2}$ at 16:00, approximately 22% higher than $Q_{EC}$. It indicates that using $Q_{EC}$ for the anthropogenic heat flux from buildings (e.g. Heiple and Sailor, 2008) may underestimate the effect of $Q_{F,B}$ on urban atmospheric processes especially during the late afternoon/early evening. In addition, $Q_{F,B}$ is always smaller than $Q_{Waste,o}$ because of the negative $\Delta L_{\uparrow,o-uo}$ and $\Delta H_{o-uo}$ causing a cooler exterior surface. This suggests using $Q_{Waste,o}$ as $Q_{F,B}$ (e.g. Chow et al., 2014) may overestimate $Q_{F,B}$ in summer.

However, in winter, mechanical heating and thermal mass effect shape the temporal pattern of $Q_{F,B}$ (Fig. 1i). The cool outdoor air temperature before sunrise results in a substantial heating load and peak $Q_{EC}$ (16.43 W m$^{-2}$ for median line) at 08:00. This heat is stored in building fabric (positive $\Delta S_{o-uo}$) and have a relatively stable release through convection and longwave radiation. Therefore the diurnal profile $Q_{F,B}$ is rather flatter and $\Delta S_{o-uo}$ has a highly consistent temporal pattern to $Q_{EC}$.


Overall, this analysis recognizes the crucial role of $\Delta S_{o-uo}$ in distinguishing $Q_{F,B}$ from $Q_{EC}$, which is highly
dependent on HVAC operation and natural ventilation (i.e., human activity of opening window). These two
factors can rapidly increase or decrease $Q_{F,B}$ while convection and longwave radiation cannot. Whereas in
winter, the larger IQR (shading) of $Q_{F,B}$ than $Q_{EC}$ indicates more day-to-day variation in $Q_{F,B}$ diurnal profile
than $Q_{EC}$. Estimates of $Q_{F,B}$ using satellite remote sensing found heat storage plays an important role in
moderating energy use within buildings (Yu et al., 2021). As the storage heat flux change modifies the diurnal
sensible heat flux pattern it modifies the surface temperature increment ($Q_{F,B}$ in remote sensing approach) and
hence the apparent energy consumption.
The diurnal profiles of $\Delta S_{o-uo}$ are not identical between seasons as people use different actions to achieve
thermal comfort in different weather conditions. This suggests the $Q_{F,B}$ and $Q_{EC}$ differences may vary between
climates and with cultural practices. In inventory methods the diurnal profiles may be limited (e.g. LUCY (Allen
et al., 2011), weekday/weekend by country) and ignore seasonal variations. However, $\Delta S_{o-uo}$ behaviour types
classes may benefit from distinguishing diurnal variation for different climates.
**3.2 Impact of different building operation modes on seasonal and diurnal variations**
Fig. 2 illustrates the impact of different building operation modes (Table 1: *ov1, ov2, ov3;* cf. *us*) on the $Q_{F,B}$
diurnal profiles. It suggests the different ventilation strategies and HVAC systems do change $Q_{F,B}$ in both
temporal pattern and magnitude, but their impacts vary among seasons.
In spring and autumn, different natural ventilation control strategies completely modify the $Q_{F,B}$ diurnal
profile, whereas HVAC system only increases the peak $Q_{F,B}$ slightly in autumn (Fig. 2i). The distinctly different
(opposite) trend in diurnal $Q_{F,B}$ pattern for *ov1* cf. *ov2* or *ov3* (Fig. 2a-c, g-i) is largely explained by the diurnal
change of $\Delta BAE_{o-uo}$ in the three cases. In *ov1* (window open, no control) the minimum outdoor air temperature
before sunrise creates the maximum indoor and outdoor air temperature difference, therefore the highest
$\Delta BAE_{o-uo}$ and peak $Q_{F,B}$ at 06:00 (30 W m$^{-2}$ for the median in Fig. 2a). Whereas *ov2* and *ov3* have the window
closed at night and early morning to avoid overcooling, therefore the minimum $Q_{F,B}$ in the early morning
(07:00). As outdoor air temperature increases through the day, $Q_{F,B}$ follows the reduced $\Delta BAE_{o-uo}$ in *ov1*,
whereas natural ventilation is active in *ov2* and *ov3*, leads to an increase in $\Delta BAE_{o-uo}$ and $Q_{F,B}$. Unlike *ov2*, *ov3*
has a clear peak (16 W m$^{-2}$ median, Fig. 2i) at 15:00, because when natural ventilation alone cannot satisfy
thermal comfort and *ov3* air conditioning is activated. But their overall patterns (IQR) are very consistent,
indicating afternoon use of air conditioning could increase $Q_{F,B}$ magnitude but have a limited impact on other





parts of the diurnal pattern. Surprisingly, negative $Q_{F,B}$ occurs around 17:00 in spring (Fig. 2a), suggesting the
occupied building has less heat emissions than unoccupied building. Because the natural ventilation at night and
morning cools down the building and reduced fabric exterior surface temperature leads to a large reduction in
longwave radiation and convection ($\Delta L_{\uparrow,o-uo}$ and $\Delta H_{o-uo}$) than increase in heat emission through natural
ventilation ($\Delta BAE_{o-uo}$) in afternoon. And the reduced overall emissions are converted into increase in storage
heat flux ($\Delta S_{o-uo}$). Negative $Q_{F,B}$ also occurs when unoccupied building is always ventilated (uv) and occupied
building is ventilated with control (*ov2* and *ov3*) in spring (e.g. Fig. B6b-c). The window is closed to avoid
excessive cooling at night in *ov2*. With $\Delta BAE_{o-uo}$ negative in this case, its magnitude is much larger than
increase in longwave radiation and convection ($\Delta L_{\uparrow,o-uo}$ and $\Delta H_{o-uo}$). The minimum $Q_{F,B}$ frequently
corresponds to the peak $\Delta S_{o-uo}$ .
In summer, *ov2* window is open most of the time (as in *ov1*) for thermal comfort, therefore the $Q_{F,B}$ has no
apparent difference to *ov1*. However for *ov3*, as air conditioning runs from morning to late night and there is a
very different diurnal profile (cf. *ov2* and *ov1*). Air conditioner use contributes to a much larger $Q_{F,B}$ (cf. *ov2*)
from 12:00 to 21:00. Not only is extra energy consumed, but it also removes heat from building to the
atmosphere in this period. In contrast, using natural ventilation as a cooling strategy (*ov1* and *ov2*) contributes to
a high $Q_{F,B}$ at night and early morning but very low even negative extra heat emission in afternoon.
Consistent with results in the other seasons, different ventilation control strategies in winter cause a large
change in $Q_{F,B}$ profile between *ov1* and *ov2*. However, the temporal pattern of $Q_{F,B}$ (IQR) in *ov2* is quite similar
to *ov3* because the supplied heat from mechanical heating system does not immediately enhance $Q_{F,B}$ with
closed window. *ov2* is the only scenario that has similar $Q_{F,B}$ and $Q_{EC}$ through the whole day. Comparison using
an unoccupied ventilated (*uv*) baseline (Fig. B.6) (cf. *us* Fig. 2) show that although $Q_{F,B}$ profiles differ, the
impacts of different building operation modes are consistent when the same occupied buildings used. The
impact of baselines with different air exchange on $Q_{F,B}$ are analysed in Sect. 3.3.



Figure 2 : As Figure 1c, f, i, j , but  comparing three different building operation types (a, d, g, j) *ov1*: window is always open

without control, no HVAC; (b, e, h, k) *ov2*: controlled natural ventilation for indoor thermal comfort, no HVAC; (c, f, i, l)

*ov3*: mixed mode ventilation



### 3.3 Impact of unoccupied baseline chosen

Here two unoccupied baselines (*us* - unoccupied sealed building, *uv* - unoccupied ventilated building with
uncontrolled open window) are used to assess the impact. A ratio between $Q_{F,B}$ to $Q_{EC}$ ($R$) is used (Fig. 3) to
normalize the impact of baselines on their difference with different building operation modes. The largest
difference in $R$ occurs on 23 December at 11:00, with values of 5.13 (*ov3-uv*) and -2.72 (*ov1-us*), reflecting the
considerable difference between $Q_{F,B}$ to $Q_{EC}$.

Two diurnal patterns of the $R$ ratio are distinguished. When the window is always open (*ov1* in all seasons,

*ov2* in summer), $R > 1$ ($Q_{F,B} > Q_{EC}$) at night/early morning (22:00-08:00), reaching its maximum around
05:00-07:00 (near sunrise in all seasons). For the remaining periods, which are relatively warm, $R < 1$. Whereas,
when window opening/closing is controlled and HVAC is used for thermal comfort an almost inverse temporal
pattern of R occurs, with R > 1 during afternoon when either window is open or the air conditioner is activated.
The peak $R$ occurs at 15:00 when both outdoor temperature and solar radiation are high.

When different unoccupied baselines are used, the temporal patterns of $R$ are similar for all cases, but their

magnitudes differ significantly. $R$ is close to 1 when window states between unoccupied and occupied buildings
are similar (e.g. *ov1-uv* in all seasons, *ov2-uv* in summer). Hence, greater difference occurs in heat transfer from
ventilation or mechanical heating/cooling between occupied and unoccupied building (i.e., larger $R$). Thus, the
baseline chosen impacts the results and require appropriate consideration for incorporating $Q_{F,B}$ into
atmospheric modelling.



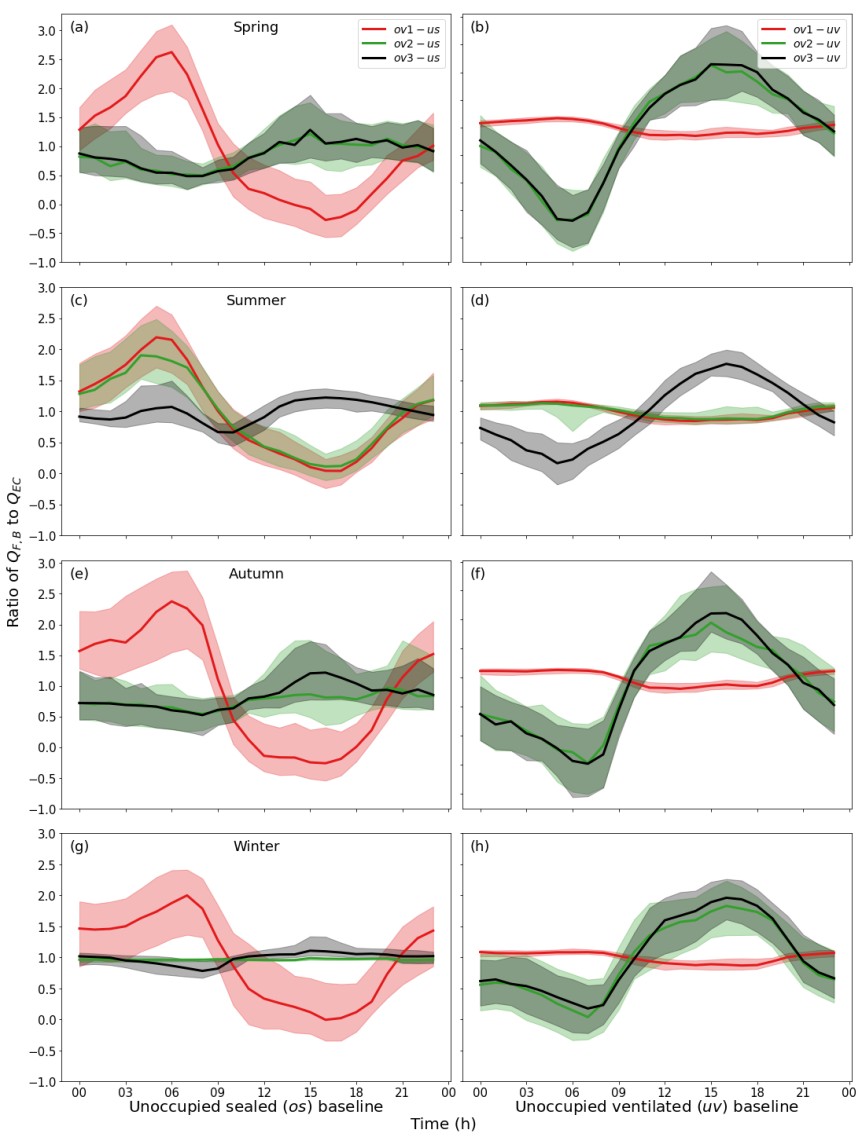

Figure 3: $Q_{F,B}$ to $Q_{EC}$ ratio (R) median (line) and IQR (shading) for (a-b) spring, (c-d) summer, (e-f) autumn and (g-h) winter, using two unoccupied baselines: (a, c ,e, g) sealed (us), and (b, d, f, h) ventilation (uv); each with three occupancy types (colour): ov1: Only internal heat gains are applied and window is fully open; ov2: Internal heat gains and natural ventilation control are applied. ov3: Internal heat gains, natural ventilation control and HVAC system are applied.

## 3.4 Daily variation of fluxes in relation to meteorological conditions

Ambient air temperature is one of the most crucial factors controlling building energy consumption (Sailor and Vasireddy, 2006). Hence, it is often used to determine daily variability of $Q_{EC}$ (e.g. Lindberg et al., 2013) and the resulting monthly variations (e.g. Allen et al., 2011). By accounting for $\Delta S_{o-uo}$ in this study, the response of





$Q_{F,B}$ to ambient air temperature may differ to previous studies. To examine this we use the *ov3-us* case to
consider the relations of daily mean (unless indicated) variables of air temperature (mean) , solar radiation (daily
total) and simulated available energy to the building from human activities ($\Delta B$) with anthropogenic heat flux
($Q_{F,B}$ in Fig. 4a), energy consumption ($Q_{EC}$ in Fig. 4 b) and their difference ($\Delta S_{o-uo}$ in Fig. 4 c).The overall
trends between $Q_{F,B}$ and $Q_{EC}$ to ambient air temperature are consistent, with $Q_{F,B}$ and $Q_{EC}$ smallest when
temperatures are between 10-15℃. This coincides with the Nicol and Humphreys'(2002) monthly balance-point
temperature of 12℃, which has been regarded as the equivalent ambient air temperature with the minimum
energy use within the building (e.g. Allen et al., 2011, Koralegedara et al., 2016). As the temperature increases
(decreases), $Q_{EC}$ increases proportionally with temperature due to mechanical cooling (heating). However, in
contrast to $Q_{EC}$, $Q_{F,B}$ has a much larger variability at the same temperature caused by a large range of $\Delta S_{o-uo}$ (-
7.7 to 9.0 W m$^{-2}$), which is highly dependent on human activities on diurnal scale (Sect. 3.1)
To understand the large daily variability of $\Delta S_{o-uo}$, we use $\Delta B$ to indicate the effect of human activities
(heat addition or removal) in one day. Higher $\Delta B$ (larger circles) are associated with higher $\Delta S_{o-uo}$ at the same
ambient air temperature, especially in winter (Fig. 4c). This is not unexpected as buildings will absorb more heat
when extra internal energy is added into the building. Inversely, negative $\Delta B$ (small circles) contributes to much
more heat release from heat storage (lower $\Delta S_{o-uo}$ through either natural ventilation or mechanical cooling. The
sign and magnitude of $\Delta B$ are linked to daily cumulative solar radiation. At the same ambient air temperature,
higher solar radiation indicates the need for larger heat removal or less heat addition to the building for thermal
comfort, therefore leading to a smaller $\Delta B$ and lower $\Delta S_{o-uo}$. Consequently, we can conclude that both ambient
air temperature and cumulative solar radiation are important meteorological factors to determining $\Delta S_{o-uo}$ and
$Q_{F,B}$.





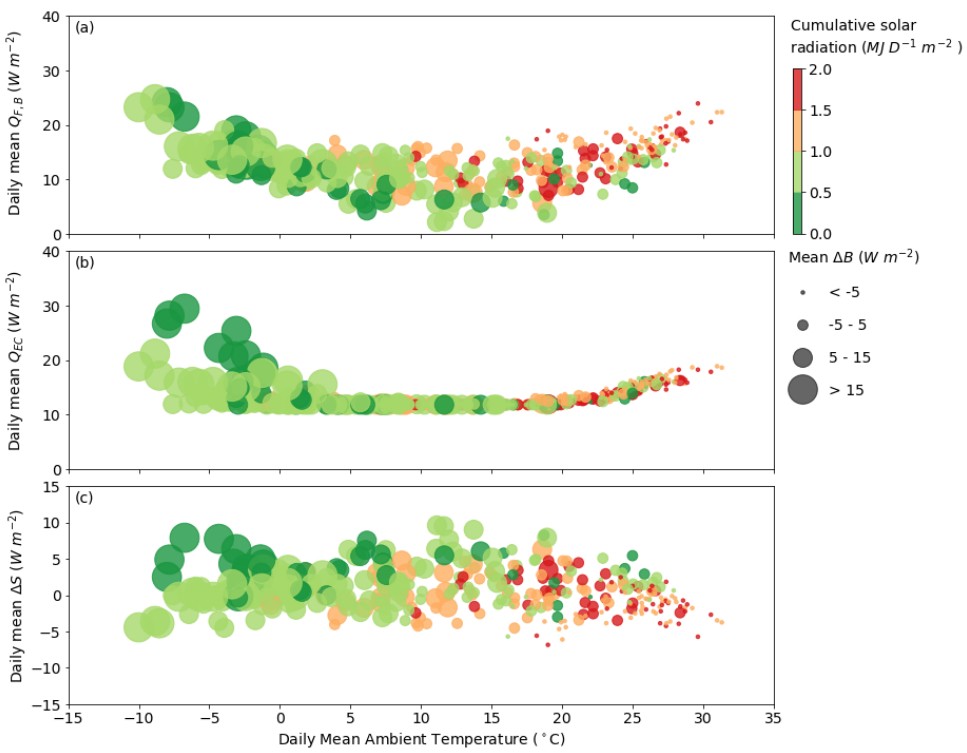


Figure 4: Daily results for the *ov3-us* case stratified by daily cumulative solar radiation (colour) and daily mean available
energy to the building (size) (Eq. (9) associated with human activities, with mean external air (ambient) temperature and (a)
mean anthropogenic heat flux, (b) energy consumption and (c) difference in storage heat flux.
**4 Conclusions**
Anthropogenic heat flux from buildings ($Q_{F,B}$) is defined as the additional heat released from building into
atmosphere due to human activities. It is qualitatively different to building energy consumption ($Q_{EC}$) in
temporal pattern and magnitude as result of thermal inertia of building (Iamarino et al., 2012). However, as there
is no standard to quantify 'real' $Q_{F,B}$ most studies use $Q_{EC}$ as a proxy via inventory and building energy
modelling approaches. This paper proposes a new method to quantify a more appropriate $Q_{F,B}$ by utilising the
difference in heat fluxes between an occupied and unoccupied building (i.e. the built structure with absolutely
no energy use and human metabolism). We show the difference between $Q_{EC}$ and $Q_{F,B}$ is attributable to a
change in the storage heat flux induced by human activities ($\Delta S_{o-uo}$). $Q_{F,B}$ has four components based on its
dissipation pathways, including outgoing longwave radiation, turbulent sensible heat flux (convection), heat
release due to air exchange and waste heat from HVAC systems. We use one simplified case study in Beijing to





demonstrate the analysis using building energy simulations to quantify the temporal difference between $Q_{EC}$ and
$Q_{F,B}$ and understand the relative importance of building operations for thermal comfort and meteorological
condition on $Q_{F,B}$. The key conclusions are:

- Hourly ratios between $Q_{F,B}$ and $Q_{EC}$ can differ between -2.72 and 5.13 because of differences in occupancy use of the building (within a year, in Beijing's climate). Individual ratios frequently exceed 3 between 14:00 and 16:00 when controlled natural ventilation or mechanical cooling is activated in shoulder season). Thus, the definitions differences are large.

- Natural ventilation ($\Delta BAE_{o-uo}$) or HVAC operation ($Q_{Waste,o}$ for cooling and $Q_{HVAC}$ for heating) are two predominant contributors to the storage heat flux. Hence, different building operations to control thermal comfort determine the diurnal profile of $Q_{F,B}$ by affecting not only $Q_{EC}$ but also $\Delta S_{o-uo}$.

- The day-to-day variation of $Q_{F,B}$ diurnal profile is broader than that of $Q_{EC}$.

- Diurnal profile of $\Delta S_{o-uo}$ varies with season as occupants modify their behaviours and the interaction with buildings to achieve thermal comfort (e.g. cooling in summer and heating in winter), indicating differences between $Q_{F,B}$ and $Q_{EC}$ will vary with both climate and cultural norms.

- $Q_{F,B}$ is sensitive to the unoccupied baseline chosen (here two are analysed unoccupied sealed vs unoccupied ventilated). An 'unoccupied baseline' needs to be integrated into urban climate modelling in the future.

- Daily mean temperature only accounts for the day-to-day variability in $Q_{EC}$ rather than $\Delta S_{o-uo}$. Both ambient air temperature and cumulative solar radiation are important meteorological factors to determine $\Delta S_{o-uo}$ and $Q_{F,B}$.

Our new approach should be used to provide data for future parameterisations of both anthropogenic heat
flux from buildings and storage heat fluxes for urban weather and climate modelling. We conclude that storage
heat fluxes in cities could also be modified by occupant behaviour. This theoretical analysis is the first step
towards a quantitative understanding on how $Q_{F,B}$ differs from $Q_{EC}$. Future work should include: (i) Expand
beyond our very idealised building archetype and building operation mode, to more complex real-world building
types and building operations; (ii) we ignore latent heat release by HVAC system, such as cooling towers, these
processes need to be included; and (iii) a wider range of building thermal properties should be explored.
**Appendix A: Building energy simulation details**
**Table A.1:** Thermal properties of building fabric material (ASHRAE, 2017)





| Opaque fabric | | | | | | |
| --- | --- | --- | --- | --- | --- | --- |
| Elements | Thermal conductivity (W m$^{-1}$K$^{-1}$) | Thickness (m) | U-value (W m$^{-2}$K$^{-1}$) | Thermal resistance (m$^2$K$^1$W$^{-1}$) | Density (kg m$^{-3}$) | Specific heat (J kg$^{-1}$K$^{-1}$) |
| *Exterior wall (inside to outdoors)* | | | | | | |
| Interior surface coefficient | | | 8.290 | 0.121 | | |
| Concrete block | 0.510 | 0.100 | 5.100 | 0.196 | 1400 | 1000 |
| Foam insulation | 0.040 | 0.0615 | 0.651 | 1.537 | 10 | 1400 |
| Wood siding | 0.140 | 0.009 | 15.556 | 0.064 | 530 | 900 |
| Exterior surface coefficient | | | 29.300 | 0.034 | | |
| Overall, air to air | | | 0.512 | 1.952 | | |
| *Floor (inside to outdoors)* | | | | | | |
| Interior surface coefficient | | | 8.290 | 0.121 | | |
| Concrete slab | 1.130 | 0.08 | 14.125 | 0.071 | 1400 | 1000 |
| Insulation | 0.040 | 1.007 | 0.040 | 25175 | 0 | 0 |
| Overall, air to air | | | 0.039 | 25.366 | | |
| *Exterior roof (inside to outdoors)* | | | | | | |
| Interior surface coefficient | | | 8.290 | 0.121 | | |
| Plasterboard | 0.160 | 0.010 | 16.000 | 0.063 | 950 | 840 |
| Fiberglass quilt | 0.040 | 0.1118 | 0.358 | 2.794 | 12 | 840 |
| Roof deck | 0.140 | 0.019 | 7.368 | 0.136 | 536 | 900 |
| Exterior surface coefficient | | | 29.300 | 0.034 | | |
| Overall, air to air | | | 0.318 | 3.147 | | |
| *Transparent fabric (windows)* | | | | | | |
| Number of panes | | | 2 | | | |
| Pane thickness (mm) | | | 3.175 | | | |
| Air-gap thickness (mm) | | | 13 | | | |
| Normal direct-beam transmittance through one pane | | | 0.86156 | | | |
| Thermal Conductivity of glass (W m$^{-1}$K$^{-1}$) | | | 1.06 | | | |
| Exterior combined surface coefficient (W m$^{-2}$K$^{-1}$) | | | 21.00 | | | |
| Interior combined surface coefficient (W m$^{-2}$K$^{-1}$) | | | 8.29 | | | |
| U-value from interior air to ambient air (W m$^{-2}$K$^{-1}$) | | | 3.0 | | | |
| Double-pane solar heat gain coefficient at normal incidence | | | 0.789 | | | |

**Figure A.1:** Building geometry of ASHRAE 140 case 900 (with changed window)

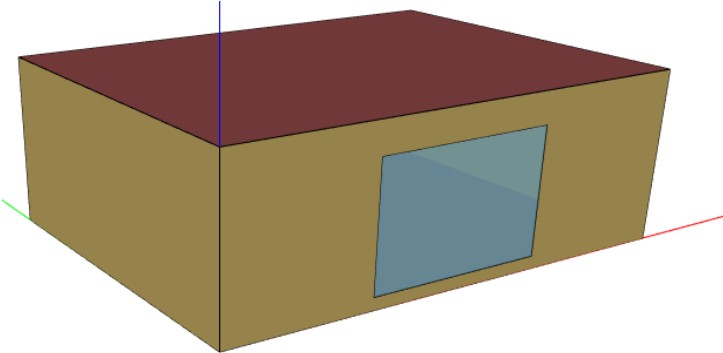


**Table A.2:** Composition of internal heat gains from local building code (MOHURD, 2018). Human metabolism rate (100 W
p$^{-1}$) is typical of resting activities (e.g. sleeping, reclining, seated and standing, 72-126W p$^{-1}$) (ASHRAE, 2005).

| Lighting (W m$^{-2}$) | Equipment (W m$^{-2}$) | Occupancy density (p m$^{-2}$) |
| --- | --- | --- |
| 5 | 3.8 | 0.03 |




**Table A.3**: EnergyPlus output variables are used here in the following equations first. $A_{Floor}$ – is total area of floor of the building ($m^2$)

| EnergyPlus output variable (Units: W) | Notation | Building volume energy balance fluxes calculated (W m$^{-2}$) | Equation (Units: W m$^{-2}$) |
|---|---|---|---|
| Outside face net thermal radiation heat gain rate | $l_\downarrow - l_\uparrow$ | Net longwave radiation | $L_\uparrow - L_\downarrow = \sum_{i=1}^{N_{surface}}(l_\uparrow - l_\downarrow)/A_{floor}$ |
| Zone total internal total heating rate | $q_{Internal}$ | Internal heat gains within the whole building | $Q_{Internal} = \sum_{i=1}^{N_{zone}} q_{Internal} /A_{floor}$ |
| Surface outside face convection heat gain rate | $q_H$ | Turbulent sensible heat flux | $Q_H = -\sum_{i=1}^{N_{surface}} q_H /A_{floor}$ |
| Zone air heat balance air energy storage rate | $\Delta q_{S.a}$ | Net storage heat flux for the building volume | $\Delta Q_S = (\sum_{i=1}^{N_{zone}} \Delta q_{S,a} + \sum_{i=1}^{N_{surface}} \Delta q_{S,s})/ A_{floor}$ |
| Surface heat storage rate | $\Delta q_{S.s}$ | | |
| AFN (Airflow network) zone exfiltration sensible heat transfer rate | $\Delta q_{BAE}$ | Heat transfer by air exchange between building and atmosphere | $Q_{BAE} = \sum_{i=1}^{N_{zone}} q_{BAE} /A_{floor}$ |
| Zone ideal loads supply air sensible heating rate | $\Delta q_{HS}$ | Sensible heating load | $Q_{HS} = \sum_{i=1}^{N_{zone}} q_{HS} /A_{floor}$ |
| Zone ideal loads supply air sensible cooling rate | $\Delta q_{AC}$ | Sensible cooling load | $Q_{AC} = \sum_{i=1}^{N_{zone}} q_{AC} /A_{floor}$ |





 **Appendix B: Energy balance analysis for other cases**


Figure B1. As Figure 1, but uses *ov1* for occupied building case in (b, e, h, k) and the heat flux difference with respect to
unoccupied sealed building (*ov1-us*) in (c, f, i, l)


Figure B2. As Figure B1, but uses *ov2* for occupied building case in (b, e, h, k) and the heat flux difference with respect to

unoccupied sealed building (*ov2-us*) in (c, f, i, l)




Figure B3. As Figure B2, but uses unoccupied ventilation baseline (a, d, g, j) and occupied building case *ov1* in (b, e, h, k)

and their difference (*ov1-uv*) in (c, f, i, l)






Figure B4. As Figure B3, but uses occupied building case *ov2* in (b, e, h, k) and their difference (*ov2-uv*) in (c, f, i, l)





Figure B5. As Figure B3, but with *ov3* in (b, e, h, k) and their difference (*ov3-uv*) in (c, f, i, l)





Figure B6. As Figure 2 but with *uv* as the baseline



**Acknowledgements**

This work is funded as part of NERC-COSMA project (NE/S005889/1), ERC urbisphere (855005) and Newton

Fund/Met Office CSSP China Next Generation Cities (SG, ZL)

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
