# Peer review of "Revising the definition of anthropogenic heat flux from"

_Atmospheric Chemistry and Physics, 2021_

## Author Comment (AC1)

We thank reviewer#1 for their comments on our paper.
- Our responses are given in RED.
- Green indicate changes to be made to the text.
- Line numbers are as per the original paper

Anonymous referee #1

The paper deals with a significant topic of anthropogenic heat from buildings. Unfortunately the authors were not aware of recent developments (e.g., the capability of EnergyPlus to calculate heat emissions from buildings in version 9.1released in 2019 ) and directly relevent publications which are not mentioned at all in the manuscript, such as:

Modeling and Analysis of Heat Emissions from Buildings to Ambient Air. Applied Energy, 2020. https://doi.org/10.1016/j.apenergy.2020.115566

A Simulation-Based Assessment of Technologies to Reduce Heat Emissions from Buildings. Building and Environment, 2021. https://doi.org/10.1016/j.buildenv.2021.107772

EnergyPlus already can calculate in detail (at sub-hourly timestep) heat emissions from buildings to ambient air by components. There is no need to re-invent the wheel (a new definition or formulas).

The reviewer unfortunately has missed the key essence of our study. The two papers referred to by authors from Lawrence Berkeley National Laboratory calculated the heat emission from buildings using EnergyPlus, but not TRUE anthropogenic heat emission.

We have clarified this (Line 53):
*If $Q_{F,B}$ is the heat released from buildings into the atmosphere as a result of human activities inside the building (including human metabolism), when the building is completely unoccupied (e.g. no operational appliances, no people; 'ghost cities' in China (Shepard, 2015) or vacant in Dublin, Kelly and Scott 2018), $Q_{F,B}$ is zero. However, heat emission from the unoccupied building is non-zero as there is still heat exchange between building and ambient environment, as there is in other environments with large mass, such as forests (e.g. Oliphant et al. 2004), and rocks (e.g. stone forest in Wang et al. 2018). $Q_{F,B}$ differs from building heat emission (BHE) (e.g. Hong et al., 2020; Ferrando et al., 2021) as the latter refers to the absolute heat flux released from buildings to the ambient environment. Shortwave and longwave radiation can enter the unoccupied internal building space through windows and conduction through walls. This energy modifies the internal building volume, influencing storage heat flux and the other terms of the energy balance. These are not anthropogenic heat flux when the energy leaves the unoccupied building but influence the heat emissions from the building.*

*References to be added:*
- *Kelly, O and Scott, P. City vacant: Dublin's hundreds of multimillion-euro empty sites and properties. The Irish Times. https://www.irishtimes.com/news/environment/city-vacant-dublin-s-hundreds-of-multimillion-euro-empty-sites-and-properties-1.3635595, 2018*
- *Ferrando, M., Hong, T. and Causone, F. A simulation-based assessment of technologies to reduce heat emissions from buildings, Build. Environ., 195, 107772, doi:10.1016/j.buildenv.2021.107772, 2021.*
- *Hong, T., Ferrando, M., Luo, X. and Causone, F. Modeling and analysis of heat emissions from buildings to ambient air, Appl. Energy, 277, 115566, doi:10.1016/j.apenergy.2020.115566, 2020*
- *Oliphant AJ, CSB Grimmond, HN Zutter, HP Schmid, HB Su, SL Scott, B Offerle, JC Randolph, J Ehman. Heat storage & energy balance fluxes for a temperate deciduous forest Agriculture & Forest Meteorology, 126, 185-201, 2004*
- *Shepard, W. Ghost cities of China: The story of cities without people in the world's most populated country, Zed Books Ltd., 2015.*
- *Wang, K, Li, Y, Li, Y, Lin, B. Stone forest as a small-scale field model for the study of urban climate. Int J Climatol, 38: 3723– 3731,2018*

We further added one session (3.4- Line 350) to compare $Q_{F,B}$ and building heat emission (BHE).

*__3.4 Comparison between $Q_{F,B}$ and building heat emission (BHE)__*
*Comparison of building heat emissions (BHE), determined using the Hong et al. (2021) approach, to $Q_{F,B}$ (this study) for the case (ov3, in Table1), shows that the former is much larger than $Q_{F,B}$ during the day but smaller at*

*night (Fig. 4) and have different diurnal patterns. Convection from the exterior envelope ($Q_H$, Figure 1b, e, h, k) is the main contributor to BHE, therefore influences the BHE diurnal pattern in each season. During the day, solar radiation is large and a major control, whereas $Q_{F,B}$ is relatively small and consistent but modified by building-human interactions (e.g., opening windows, activation of mechanical heating and cooling systems). In the scenario, natural ventilation and mechanical cooling dominate $Q_{F,B}$ in summer and shoulder season; while in winter in their absence convection and longwave radiation are more important.*

[Figure]

*Figure 4 Comparison of seasonal diurnal $Q_{F,B}$ (ov3-us) and BHE (ov3) for (a) spring, (b) summer, (c) autumn and (d) winter.*

The use of the shoebox model to simulate and analyze the building heat emission is questionable as the shoebox model is not a real building. In ASHRAE Standard 140 context, the building model is used as a common simplified benchmark model to test and compare results from various building energy modeling tools. Why use this model? If the analysis is to study how human activities influence building operation and thus heat emissions, a real residential or commercial office building model should be used.

As we propose a new approach to calculate the anthropogenic heat from buildings. We use a well-validated simple 'shoebox' case to generate fundamental insight on how human activities (i.e. open/close windows, space heating/cooling) affect the temporal variation of anthropogenic heat and change of storage heat flux. Use of office/residential building model could provide more realistic result of anthropogenic heat profile, but also bring other layers of uncertainty and difficult to generalize. Our goal, in this paper, is not to calculate a particular anthropogenic heat flux for a particular condition - but to demonstrate the validity of the approach.

The manuscript has some typos, e.g., "and time lag and are poorly quantified"

Thank you for spotting this. We will revise accordingly and proof-read the whole manuscript to correct any other possible typos.

---

## Author Response (AR1)

We thank the reviewers for their helpful comments on our paper.

- Our responses are given in Red.
- *Green italic* indicate changes to be made to the text.
- Line numbers are as revised version with mark-up

Reviewer: Qi Li

This paper has provided an innovative perspective on the definition of anthropogenic heat flux QF,B, which distinguishes it from the widely used proxy building energy consumption QEC. The key connection between them lies in the building storage heat flux. The paper started with a conceptual discussion of their distinctions and then conducted building energy simulation using EnergyPlus for a typical building in Beijing, China. Based on a one-year simulation, the building storage heat flux and other relevant surface energy budget terms are compared and discussed for different scenarios of building energy use and differences between QEC and QF,B. The results caution the effects of such differences in future studies using the inventory approach to quantify anthropogenic heat flux, especially in atmospheric modeling. The paper is well written and organized. I have only minor comments.

Comments:

1. line 43: The last sentence of this paragraph seems to only discuss the building energy used for heating. It can be rephrased in a more general heating and cooling context.

The heat generated inside buildings by human activities could occur independent of whether a building is heated or cooled. We modify the statement to more explicitly indicate this.

**Line 43:**

*For example, the heat generated from human activities inside buildings is released initially indoors (via heating or cooling application), then transported through the building fabric by conduction, allowing it to be transported into atmosphere by turbulent sensible heat flux and outgoing longwave radiation. In this process the net storage heat flux ($\Delta Q_S$) of building is modified since building fabric temperature is changed by absorbing more heat from the internal heat generation.*

2. line 53 – 57: The first two sentences appear a bit contradicting. The first sentence says that QF,B is the heat released from buildings into the atmosphere and is zero when building is unoccupied. The second sentence says that 'heat released from the unoccupied building is non-zero'. I understand that heat released from the unoccupied building is not QF,B. Perhaps it is less confusing and more consistent with introduction of Eq. 1 and Eq. 5 to mention first that regardless of whether human activities are present, buildings exchange (as you showed later QF,B can be positive or negative, so not just releasing heat) heat with the atmosphere

To more clearly differentiate these two terms, i.e., building anthropogenic heat and building heat emission we have modified the text.

**Line 59:**

*If $Q_{F,B}$ is the additional heat released from buildings into the atmosphere as a result of human activities inside the building (including human metabolism), then when the building is completely unoccupied (e.g. no operational appliances, no people: such as 'ghost cities' in China (Shepard, 2015) or vacant in Dublin (Kelly and Scott, 2018)); then $Q_{F,B}$ is zero. However, the heat released from the unoccupied building is not zero because there is still heat exchange between the building and ambient environment (see Eq. 1 and 2) as occurs in other environments with large mass, such as forests (e.g. Oliphant et al., 2004) and rocks (e.g. Wang et al., 2018). $Q_{F,B}$ differs from building heat emission (BHE) (e.g., Hong et al., 2020; Ferrando et al., 2021) as the latter is the total heat flux released from buildings to the ambient air ($BHE_{uo} = Q_{H,uo} + Q_{BAE,uo} + L_{\downarrow,uo[air \to boi]} - L_{\uparrow,uo[boi \to air]}$) not due to human activities alone. Shortwave and longwave radiation can enter the unoccupied internal building space through windows and conduction through walls. It modifies the heat storage within building volume and temperature of building envelope and indoor air, subsequently have a further influence on heat emission through sensible heat flux, outgoing longwave radiation and air exchange. But this energy leaving the unoccupied building is not anthropogenic heat flux.*

3. Eq. 5 has subscripts 'o' to distinguish some of the common terms from those in Eq. 1. However, to emphasize, it may be helpful to re-iterate that introducing of Qinteral,o on the LHS modifies the entire energy balance in your chosen control volume.

Modified to introduce an additional time.

**Line 76:**

*For an occupied building, the internal heat gain arises from: (1) the equivalent sources and sinks as the unoccupied buildings; but also (2) the energy linked to the indoor human activities (metabolism, powered appliances and energy inputs to heating or cooling). These will modify each of the energy balance fluxes. Some of this additional energy is transported out of buildings through indoor-outdoor ventilation exchange and/or HVAC systems, immediately contributes to $Q_{F,B}$, while some is stored in the building fabric, and is later released outdoors through various pathways (convection, radiation, conduction) to become $Q_{F,B}$ with a time lag.*

4. Eq. 15: It appears to the reviewer that a long time limit exists, such that QF,B = QEC, as the long time average of storage heat flux tends to zero. Also, does the simulation data support this speculation? Maybe this is helpful to comment on, especially regarding using QEC from the inventory as a proxy for some annual/seasonal averaged quantities.

Yes, we agree. Our year-long simulation data from building energy simulation show that annual mean storage heat flux for case *us* and *ov3* is -0.026 W m$^{-2}$ and 0.068 W m$^{-2}$. Their difference is much smaller compared to daily (from -7.685 to 9.035 W m$^{-2}$) and sub-daily (from -44.848 to 41.818 W m$^{-2}$) scale. Thus it can prove $Q_{F,B}$ = $Q_{EC}$ for annual resolution. The following text is added:

**Line 163:**

*The sources of $Q_{F,B}$ are from both energy consumption ($Q_{EC}$) and difference of storage heat flux ($\Delta S_{o-uo}$) between unoccupied and occupied building ($Q_{F,B}$ in this study includes part of $Q_{F,M}$ from human metabolism).* In most prior studies, the second term of Eq. (15) is ignored. Although the storage heat flux over a year should tend to zero, over short periods (e.g. sub-daily) $\Delta S_{o-uo}$ is not zero causing time lag and magnitude difference between $Q_{F,B}$ and $Q_{EC}$. *Therefore, estimation of $Q_{F,B}$ by differences in heat emission between occupied and unoccupied building can capture the impact of dynamic changes in the building storage heat flux especially at sub-annual temporal cycle.*

5. Fig.3: it may help visually to put a R=1 line.

The line R=1 is added in updated figure

**Line 383:**

[Figure]

*Figure 1: $Q_{F,B}$ to $Q_{EC}$ ratio (R) median (line) and IQR (shading) for (a-b) spring, (c-d) summer, (e-f) autumn and (g-h) winter, using two unoccupied baselines: (a, c ,e, g) sealed (us), and (b, d, f, h) ventilation (uv); each with three occupancy types (colour): ov1: Only internal heat gains are applied and window is fully open; ov2: Internal heat gains and natural ventilation control are applied. ov3: Internal heat gains, natural ventilation control and HVAC system are applied.* Black dotted line: Ratio R=1*

6.  Line 365: does ΔB refer to the definition in Eq. 9?

Yes, ΔB is the net available energy added to or removed from the building in Eq.9. It has been cross referenced in the following text

**Line 416:**

*To understand the large daily variability of $ΔS_{o-uo}$, we use ΔB (net available energy from human activities in buildings in Eq. (9)) to indicate the effect of human activities (heat addition or removal) in one day*

7.  Does the ambient temperature considered in the building energy simulation kept as an externally imposed forcing? i.e., there is no feedback between heat released from the buildings and the ambient air

Yes, in this current study the ambient temperature in building energy simulation is from TMY forcing data without feedback of heat ejection from building. We investigate this further elsewhere.

Reviewer: Alberto Martilli
In this manuscript authors analyze the difference between Building Energy consumption and heat flux ejected to

the atmosphere by buildings due to human activities. They base their analysis on the comparison between a no occupied building and one occupied. They first show, using conservation equations, why these two quantities are different, and then they use a series of EnergyPlus simulations to quantify the differences. I like this manuscript. I think it highlights a very important point for urban atmospheric modelling. My few comments below are oriented to improve the clarity and strength the message.

Comments:

1. I strongly suggest authors to include a list of symbols. This will improve the clarity and help the readers to follow the equations.

List has been added.

**Line 512:**

*Nomenclature*

| | |
|---|---|
| $A_{eff}$ | Effective area of windows opening ($m^2$) |
| $\Delta B_{o-uo}$ | Net available energy from human activities in building ($W\ m^{-2}$) |
| $\Delta BAE_{o-uo}$ | Difference in heat transfer by air exchange between building and atmosphere between occupied (o) and unoccupied (uo) building ($W\ m^{-2}$) |
| $BHE$ | Building heat emission to ambient air ($W\ m^{-2}$) |
| $\Delta H_{o-uo}$ | Difference in $Q_H$ between occupied (o) and unoccupied (uo) building ($W\ m^{-2}$) |
| $F_{[sky \to boi]}$ | View factor from sky to building of interest |
| $F_{[other\ b \to boi]}$ | View factor from other buildings to building of interest |
| $F_{[boi \to sky]}$ | View factor from building of interest to sky |
| $F_{[boi \to other\ b]}$ | View factor from building of interest to other buildings |
| $C_d$ | Discharge coefficient |
| $H$ | Height of windows opening (m) |
| $K_\uparrow$ | Outgoing shortwave radiative flux ($W\ m^{-2}$) |
| $K_\downarrow$ | Incoming shortwave radiative flux ($W\ m^{-2}$) |
| $L_\uparrow$ | Outgoing longwave radiative flux ($W\ m^{-2}$) |
| $L_\downarrow$ | Incoming longwave radiative flux ($W\ m^{-2}$) |
| $\Delta L_{\uparrow,\ o-uo}$ | Difference in $L_\uparrow$ between an occupied (o) and unoccupied (uo) building ($W\ m^{-2}$) |
| $\Delta Q_S$ | Net storage heat flux for the building volume ($W\ m^{-2}$) |
| $Q^*$ | Net all-wave radiative flux ($W\ m^{-2}$) |
| $Q_{AC}$ | Sensible cooling load from air conditioning ($W\ m^{-2}$) |
| $Q_{BAE}$ | Heat transfer by air exchange between building and atmosphere ($W\ m^{-2}$) |
| $Q_{F,\ B}$ | Anthropogenic heat flux from building sector ($W\ m^{-2}$) |
| $Q_{F,\ M}$ | Anthropogenic heat flux from metabolic activities ($W\ m^{-2}$) |
| $Q_{F,\ T}$ | Anthropogenic heat flux from transport ($W\ m^{-2}$) |
| $Q_H$ | Turbulent sensible heat flux ($W\ m^{-2}$) |
| $Q_{HS}$ | Sensible heating load ($W\ m^{-2}$) |
| $Q_{HVAC}$ | Energy consumption by heating ventilation and air conditioning (HVAC) system ($W\ m^{-2}$) |
| $Q_{Internal}$ | Internal heat gain within the building (human metabolism, lighting and appliance) ($W\ m^{-2}$) |
| $Q_{Waste}$ | Waste heat released to outdoor by HVAC system ($W\ m^{-2}$) |
| $R$ | Ratio of anthropogenic heat flux from building ($Q_{F,\ B}$) to energy consumption ($Q_{EC}$) |
| $\Delta S_{o-uo}$ | Difference in storage heat flux between occupied (o) and unoccupied (uo) building ($W\ m^{-2}$) |
| $T_{ave}$ | Average indoor and outdoor air temperature (°C) |
| $\Delta T$ | Indoor and outdoor air temperature difference (°C) |
| $U_W$ | Reference wind speed at height of upstream airflow ($m\ s^{-1}$) |
| $V_{Stack}$ | Buoyancy driven ventilation rate ($m^3\ s^{-1}$) |
| $V_T$ | Total ventilation rate from combined wind and buoyancy effects |
| $V_W$ | Wind driven ventilation rate ($m^3\ s^{-1}$) |

2. Authors define $Q_{F,B}$ as "…the heat released from buildings into the atmosphere as a result of human activities inside the building (including human metabolism)" (lines 53-54). Then in $Q_{F,B}$ they include $Q_{waste,o}$ , the heat emitted to the atmosphere by HVAC. Indeed, running the HVAC is a result of a human activity, so it is understandable to include $Q_{waste,o}$ in $Q_{F,B}$ , but I think it would be worth reminding here that not all the $Q_{waste,o}$ is coming from heat generated inside the building by human activities (e. g. lighting, powered appliances and

metabolism). A significant part of this heat is energy that has entered in the building from outside in form of radiation through the windows, or heat diffusion through the walls. In other words, an empty building with HVAC functioning, would still have Q$_{waste,o}$ different than zero. This energy, that is not from anthropogenic origin, would have been stored in the building without HVAC. This message (to me one of the most important of the study), is already implicit in the manuscript, but I believe it should be made explicit in the text.

We agree with the reviewer. This is very important and central to our calculations -i.e. the difference between the natural heat entering a building, and being rejected outdoors by a HVAC system (human activity) vs not – would be the simplest case (i.e. with no people or additional energy use). We have expanded the text

**Line 109:**

*(2) Q$_{HVAC,o}$ : energy consumption in the building from heating, ventilation and air conditioning (HVAC) system. As the building may emit exhaust/waste heat (e.g. via HVAC systems), there is an additional sink (RHS) referred to here as Q$_{Waste,o}$.* *The cooling system, Q$_{Waste,o}$ will remove energy from both anthropogenic (e.g. metabolism, lighting, electrical appliance and Q$_{HVAC,o}$) and natural sources (e.g. solar radiation through windows, heat diffusion through building envelope). Thus, only the natural sources occur in both the occupied and unoccupied states. In a 'simple' occupied state, with HVAC operated only (i.e. no people or other appliances) there is a difference in the building storage heat flux because of the alternative route to transport this natural heat of the building out from the additional source of energy.*

3. Lines 57-58 "This energy modifies the internal building volume…". Energy cannot modify the internal building volume, Please clarify.

It has been changed to make this clearer:

**Line 67:**

*Shortwave and longwave radiation can enter the unoccupied internal building space through windows and conduction through walls.* *It modifies the heat stored within the building volume and the temperature of the building envelope and indoor air, subsequently influencing the emission of heat via sensible heat flux, outgoing longwave radiation and air exchange. But this energy leaving the unoccupied building is not anthropogenic heat flux.*

4. Lines 65-66 "Some of this additional energy is transported out of buildings through indoor-outdoor ventilation exchange and immediately contributes to Q$_{F,B}$". HVAC also would immediately eject the additional heat.

It has been added:

**Line 80:**

*Some of this additional energy is transported out of buildings through indoor-outdoor ventilation exchange and*/*or HVAC systems*, *immediately contributing to Q$_{F, B}$.*

5. Line 79 Just to improve clarity, and avoid misunderstanding, I would remind here that Q$_H$ is not the sensible heat flux that could be measured on a mast in the inertial sublayer. This should include also Q$_{Waste}$ and Q$_{BAE}$.

The following text has been added:

**Line 117:**

*Here Q$_H$ only represents the convection heat transfer at building external surface (i.e. wall, roof and windows). Both Q$_{Waste,}$ and Q$_{BAE}$ will be incorporated into the turbulent sensible heat flux by the time they reach the inertial sub-layer (ISL) or constant flux layer (CFL). Hence, sensors (e.g. eddy covariance or large aperture scintillometry) located in the ISL would observe this as Q$_H$. The separation of these three terms is to better understand how human activities (e.g. open/closed windows, HVAC operation) influence each heat flux. Urban*

*canopy parameterisation (UCP) can use this information about the separate sources and their roles in the urban energy balance to account for the modified fluxes by the time they reach the ISL. Additionally, it is clearer for multi-layer UCP where vertically the energy should enter.*

6. Figure 1. If ΔH is the difference in sensible heat flux from building surfaces between an occupied and an unoccupied building, I am a bit surprised to see such small variability between night and day. I would expect that the change in storage would affect more the heat flux, in particular during night

As clarified in comment 5, *ΔH* represents the difference in convection heat transfer at building external surface, not sensible heat flux as in urban surface energy balance.

In summer and the shoulder season, the natural ventilation (ΔBAE, Figure 1c, f, i) and mechanical cooling ($Q_{Waste}$, Figure 1f) are two dominant pathways to release $Q_{F, B}$. The transfer of generated and stored heat by those two pathways are faster than conduction through insulated envelope (U-values used are 0.512 (external wall) and 0.318 (roof)). Hence, the change in storage heat flux (*ΔS*) has an inverse diurnal pattern to ΔBAE and $Q_{Waste}$. The impact on *ΔH* is relatively small and consistent in these three seasons.

In winter, without natural ventilation and mechanical cooling (Figure 1l), *ΔH* is the main pathway to emit heat and its magnitude is higher than other seasons. As the internal heat gains are kept fixed (11.8 W m$^{-2}$), the diurnal variation of *ΔS* and *ΔH* mainly depends on spacing heating. In the heating period (4:00 to 10:00, the varying median (line) $Q_{EC}$), diurnal pattern of *ΔS* and $Q_{EC}$ are very similar, while *ΔH* is almost flat indicating most of the supplied heat is stored by building fabric and transported by conduction (insulation layer is external to concrete). At 14:00 ΔH starts to increase as ΔS becomes negative; i.e. stored heat from spacing heating is transmitted through the insulation layer to warm up external surface. The 10-h time lag is related to the thermal inertia of the building. The small peak in ΔH is attributable to a relatively high thermal resistance.

7. Based on this study – Can authors derive some recommendations to users and developers of urban canopy parametrizations that include simplified building energy models?

We add the following in conclusion section

**Line 472**

*For developers of urban canopy parameterisations (UCP) there are several considerations because of computational efficiencies essential for undertaking weather and climate modelling: (1) human activities within building are modifying both the storage heat flux and the anthropogenic heat flux; (2) assuming within an UCP that a 'simple' building energy model (BEM) (cf. a full building energy simulation scheme such as EnergyPlus) will require some human activities to be simplified, such as using fixed ventilation rate, instead of dynamic natural ventilation depending on both outdoor weather condition and thermal comfort requirements; and (3) with a multi-layer UCP the appropriate levels for the impact of these energy exchanges can be accounted for. Our current research is extending this analysis to consider moisture; and exploring the role of building materials, construction, other aspects of building design and external meteorology. The outcome of this work will also have implications for UCP development, as can help identify what can be simplified and what are critical controls in different climates and urban settings.*

8. As suggestion for future studies: it would be interesting to investigate how the use of HVAC systems modulate the temporal behavior of the storage term also for other types of buildings (e. g. commercial), that have a different human behavior (occupancy, schedule of operation, etc.). Given that storage is key in the development of the nocturnal UHI – how does HVAC affect the strength of the nocturnal UHI?

We added the following text

**Line 461:**

*Our new approach should be used to provide data for future parameterisations of both anthropogenic heat flux from buildings and storage heat fluxes for urban weather and climate modelling.* We conclude that storage heat fluxes in cities is also being modified by occupant behaviour, particularly by natural ventilation and mechanical cooling. It is expected that the diurnal variation of $\Delta S_{o-uo}$ will vary with operation schedules for different building uses (e.g. residential vs. commercial buildings). Given the release of stored heat is critical influence on the nocturnal canopy layer urban heat island (CL-UHI), the impact of different HVAC operations on nocturnal UHI should be explored further. This is an important factor to determine diurnal pattern of $Q_{F,B}$ in the shoulder season and can be expressed more accurately. However, in different climates and with different social cultural practices the periods most influenced will change. Further studies are being conducted to explore the impacts of these, while also addressing feedbacks at the neighbourhood scale.